# Using Multiple Robots to Increase Suggestion Persuasiveness in Public Space

**Marcos Inky Tae [1,*], Kohei Ogawa [2], Yuichiro Yoshikawa [1] and Hiroshi Ishiguro [1]**

1   Intelligent Robotics Laboratory, Department of Systems Innovation, Graduate School of Engineer Science, Osaka University, Osaka 565-0871, Japan; yoshikawa@irl.sys.es.osaka-u.ac.jp (Y.Y); ishiguro@irl.sys.es.osaka-u.ac.jp (H.I.)
2   Department of Information and Communication Engineering, Graduate School of Engineering, Nagoya University, Nagoya 464-8601, Japan; k-ogawa@nuee.nagoya-u.ac.jp
*   Correspondence: tae.marcos.inky@irl.sys.es.osaka-u.ac.jp; Tel.: +81-070-1424-3936

**Featured Application: A system using the "sequential persuasion" concept would be able to increase the persuasiveness of a message for public service announcements or advertisements. A potential application would be a system of robots distributed throughout an environment, for example, a hotel, that would provide not only informational service, but also make personalized suggestions for the guests.**

**Abstract:** Though existing social robots can already be used in a variety of applications, there are technical limitations to their use, especially outside the laboratory, and humans do not fully trust or recognize them. Considering these problems, a method to make humans accept a robot's suggestion more easily was investigated. An idea called "sequential persuasion" was developed to use multiple robots distant from each other to deliver small messages, rather than a single robot for the entire interaction. To experimentally validate this concept, a field experiment was performed on a university campus. Two bottles of hand sanitizer were placed in one of the entrances to a building, and their usage was observed under three different conditions: no robot, one robot, and three robots. As people passed through the entrance corridor, the robots promoted the usage of the hand sanitizers. After several days of testing, it was found that the usage increased progressively from no robot to one robot to three robots, indicating that the number of robots influenced the behavior of the humans.

**Keywords:** social robot; social behaviors; robot behavior coordination; human–robot interaction; persuasion; sequential persuasion

## 1. Introduction

Several science fiction works have portrayed a society in which humans and robots coexist. From Asimov's I, Robot [1] to Quantic Dream's Detroit: Become Human [2], it is already part of the human imagination to have robots fully integrated into society, working like other humans, and even behaving like other humans. Accordingly, much work has been carried out to achieve this goal. Studies on the very nature of human–robot interactions [3], methodologies [4], metrics [5], and psychology [6] are examples of efforts to build the foundation of the interface between humans and robots.

In numerous studies, researchers have investigated the viability of introducing robots into society in various roles. There are examples of robots delivering flyers [7], assisting in language learning [8], and advising on healthcare decisions [9]. However, even if the robots are successful in the short term, humans often eventually become bored and disappointed with them because the gap between the user expectations and their actual experience is too wide [10].

Using robots to persuade humans has already been studied from many different angles. Many different studies have shown that robots can be used in many different

scenarios, such as food recommendation [11], social assistance [12], and storytelling [13] with varying degree of persuasiveness. Much like when receiving a message from a human, we perceive messages from robots to be more or less persuasive depending on psychological and social factors around the message. There are studies showing the effects of a robot having emotional or logical persuasive strategies [14], social cues [15] and even gender [16] on the persuasiveness of their message.

The robot CommU (Figure 1), a small humanoid robot, was demonstrated to be capable of performing a wide variety of research tasks, such as cooperating and communicaing with humans [17] and guiding adolescents with autism spectrum disorder [18]. Considering the application in a public space, humans tend to trust robots less because there is no social frame attached to them, which hinders their performance [19].

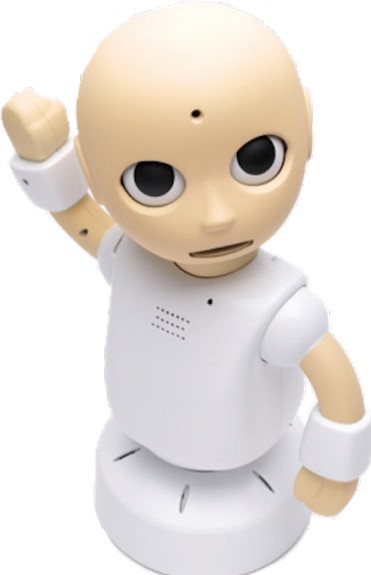

**Figure 1.** Small humanoid robot, CommU.

To counteract this problem, different strategies have been studied, such as using face-to-face contact to draw the attention of a passerby [20] to a single robot. This paper, however, proposes a new concept of using not only one, but multiple robots in a more effective configuration so that the message that the robots are to communicate is more persuasive. Through a field experiment, it was found that using multiple robots in such a configuration led to a better performing system than the standard one, indicating that this configuration can improve the persuasiveness of a social robot system in a public space, namely, the entrance corridor of a building in the university.

## 2. Sequential Persuasion

Since the objective of this study was to find a practical application in a public space, interactions with the small humanoid robots needed to be spontaneous and random, not requiring an explanation or a briefing to start. In addition, because the range of conversation necessary for an autonomous agent to reply to people is too broad to be defined easily, assisted teleoperation was used as the only practical option to control the robots.

With these limitations in mind, a well-known idea in human communication research is revisited in a novel way with multiple robots—sequential persuasion, which would entail a series of short interactions between small robots and a human, made in succession, separated by time and space, with the objective of persuading the human to engage in a specific behavior. Sequential persuasion itself is not a new word in human communication research. Giving requests in sequential ways has been regarded as persuasive for humans, for example, foot-in-the-door [21] and door-in-the-face [22] strategies. It is worth noting

that a meta-analysis on extensive studies revealed their limited effects [23]. It is also shown that the foot-in-the-door strategy is effective even in human–robot conversation [24].

When beginning an interaction, the robot would be able to introduce its purpose, the promotion of a behavior, and continue its task in the next interaction, as exemplified in Figure 2. A system so designed would be able to use both autonomous or teleoperation approaches, with a limited scope of themes and answers, thus not requiring extensive flexibility.

Such an interaction raises the question of how humans should be approached. For instance, when promoting a product to a customer, a vendor has the option to actively approach the customer or simply wait to be approached by them. Moreover, since the interaction is supposed to be short yet occur multiple times, there were options for using only one robot to convey every message or multiple robots with each responsible for one message.

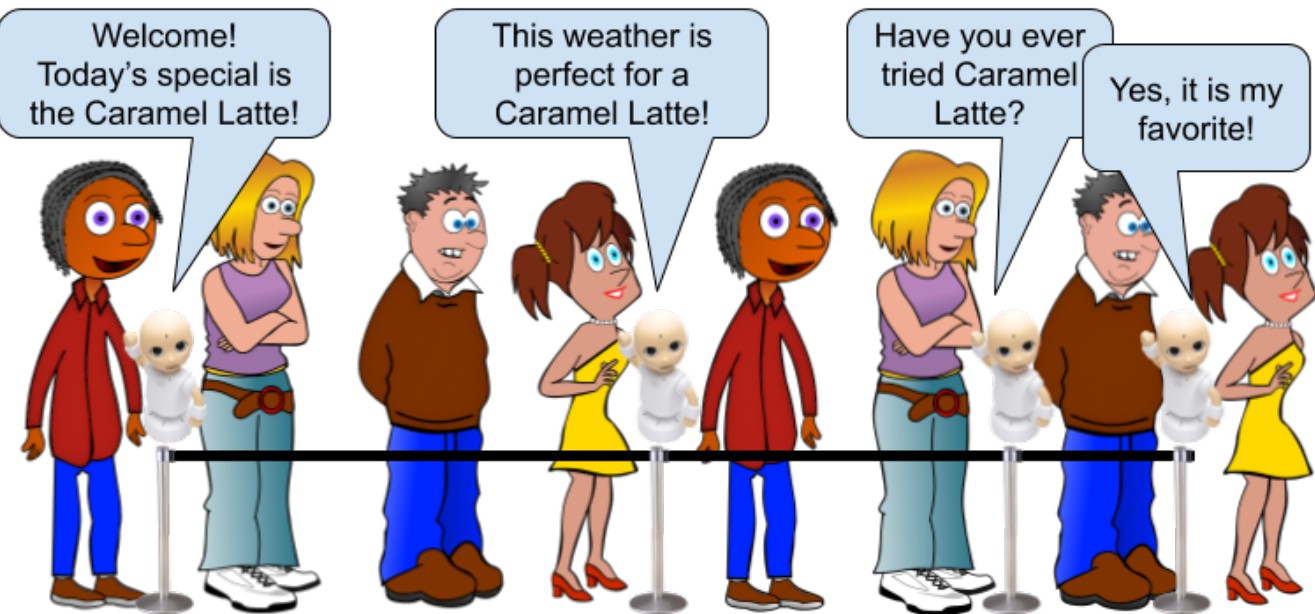

**Figure 2.** Example of possible implementation of sequential persuasion in a coffee shop line to promote a specific drink.

### 2.1. Active versus Passive

In the information age, how one perceives and consumes information is an important matter to consider when designing informative systems. Social media usage has been a frequent subject of studies, and while having the option to choose between accessing information actively or only receiving it passively was demonstrated not to affect the accuracy of one's judgment, those who passively received information were found to demonstrate greater confidence in their judgment [25].

Furthermore, studies of user engagement, performed on Italian consumers [26] and museum visitors [27], have revealed that various consumers have personality traits that resonate with different types of engagement. While campaigns that promote the expressiveness of the user are more influential on active users, "thoughtful" engagements are more suitable for passive users. To create an experience that engages as many users as possible, a system should consider ways to captivate each type of user.

Therefore, the process should not be limited to a single approach as different types of people react differently to each approach. For the sake of simplicity of the experiment, it was decided that the robot(s) would be static and would try to engage in conversation whenever possible with a human in this experiment; however, it would be possible and preferable to implement different strategies to make the robots more effective in future research.



*2.2. Single versus Multiple*

When delivering a message, it is preferable that the message lasts as long as possible with the receiver. The more persuasive a message is, the more effectively it will change the receiver's behavior.

To increase the effectiveness of the robot message, a survey of the literature on the theory of persuasion was conducted. In developing the elaboration likelihood model of persuasion [28], there are two main routes to achieve persuasion: the central route, related to thinking, and the peripheral route, related to cues that trigger feelings such as humor or desire. Furthermore, the routes are connected to the receiver's motivation and understanding of the subject.

In addition, while exploring subjects that are commonly known to promote persuasion, it was found that message repetition can have an inverted effect if the message is overexposed [29]. However, the number of information sources may have a positive effect on the persuasiveness of the message, as long as the subject is not distracted [30] and the different sources are not recognized as a single group but rather are seen as different individuals by the receiver [31].

As such, to create a persuasive message, it is desirable to create an environment that engages the receiver to ponder over the subject, that is, to persuade them on the central path, as well as providing opportunities to engage them in a peripheral manner, in the case of their lack of involvement for any reason.

Since it was determined that multiple information sources lead to greater persuasiveness if certain conditions are achieved, such as the perception that each source has its own individual thought and is not part of the same larger group, it should be considered whether a human would perceive a system of robots as a group of multiple robots that can be treated as independent agents. If it is true, the influence of the messages to request a certain behavior can be enhanced by being produced from multiple robots.

However, using multiple robots also has the advantage of making the use of passive socialization possible, in which the robots may create a conversation between the human and themselves, even though the robots are not aware of human speech [32].

## 3. Materials and Methods

To test whether the concept was valid, a field experiment was proposed. The objective of this experiment was to observe whether changing only the number of robots and not changing the message would result in a change in behavior, namely the usage of hand sanitizer by passers-by. On alternating days, different numbers of robots were placed, and the change in behavior of the subjects was observed.

*3.1. Materials*

In order to perform this experiment, two hand sanitizer dispensers, three robots, and three wooden shelves to support the robots were placed. Five video cameras were positioned to monitor passers-by, one below each robot to capture the participant's foot and one on top of each hand sanitizer dispenser. Since it was possible to affect participants' behavior, the angles of them were carefully chosen not to capture participants' faces to minimize the influence of the participant's behavior. One laptop was set up to act as the server, with four Ethernet cables and a switch to connect all the robots to the server, and a Wi-Fi router and a smartphone connected to the Wi-Fi to send commands to the server. The network system configuration is shown in Figure 3.

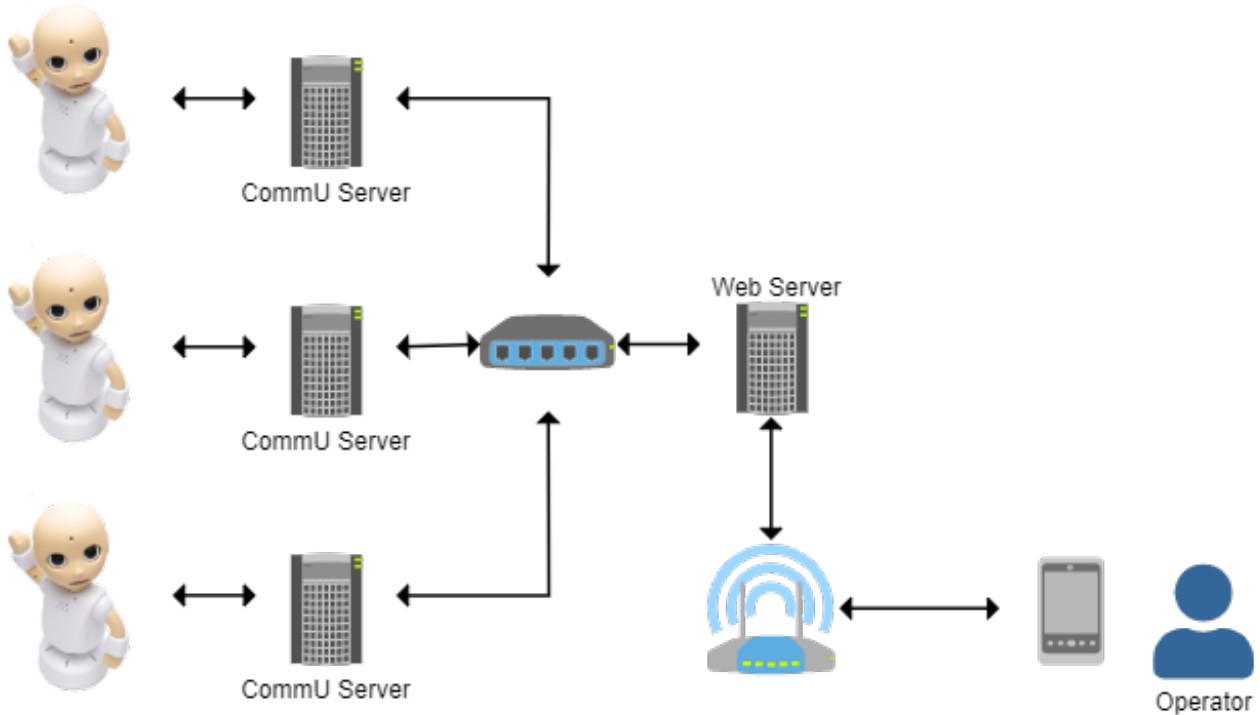

**Figure 3.** Network system configuration used for the field experiment.

### 3.1.1. CommU

CommU is an interactive robot that allows humans to experience a higher level of interaction by allowing multiple robots to interact with each other. It is equipped with cameras, microphones, speakers, voice synthesizers for both Japanese and English, and network functions. The head has eight degrees of freedom to allow the robot to move its eyes, open and close its mouth, while each arm and its waist have two degrees of freedom, respectively, to produce other bodily movement such as gestures.

### 3.1.2. System Description

To control the robots, the operator used his smartphone to access a website that contained several buttons to both activate the robots and make notes on the participants. The website was hosted on a physical web server that was placed on-site and was developed using Python with the Django framework. To access the website, both the web server and the smartphone were connected to a Wi-Fi router.

As shown in Figure 4, the operator had buttons, C1, C2, and C3, to activate each robot (CommU1, CommU2, and CommU3, respectively) and buttons, ABY, ACY, BAY, BCY, CAY, CBY, ABN, ACN, BAN, BCN, CAN, and CBN, to make notations on participants' behavior. Each letter represents how the participant entered the experiment area, how they left, and whether they used the hand sanitizer. For example, ABY indicates that a person entered from area A (the outside door), left through area B (the corridors), and used the hand sanitizer (Y = Yes), while CAN indicates that a person entered from area C (the stairs), left through area A (the outside door), and did not use the hand sanitizer (N = No). Every time a button was pressed, a log was registered in the web server internal database, and the counter below the button increased. Additionally, there was a button to change the language that the robot spoke between English and Japanese.

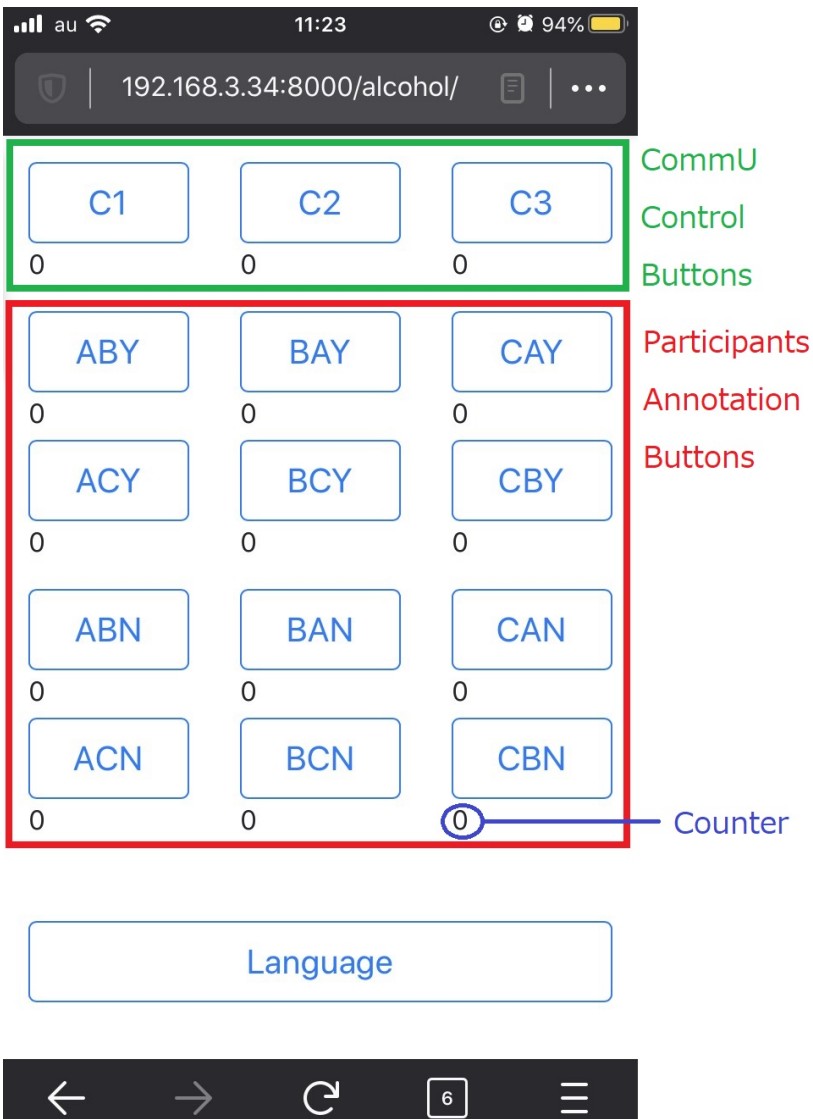

**Figure 4.** User interface of the website to activate robots and make annotations on participants' routes on the smartphone. As annotated in the image, the buttons C1, C2, and C3 activate the robots, while other buttons were used to record the participants' routes and whether they used the hand sanitizer. Below each button is a counter to show how many times each button was pressed.

When the button to activate a robot was pressed, an HTTP request was fired from the web server to the corresponding CommU internal server to activate the robot and deliver its message. The web server was connected to each CommU internal server through Ethernet cables connected by a switch. The message that the CommU delivered was a random phrase from the following list:

- "Don't forget the sanitizer!"
- "Did you clean your hands?"
- "Hand sanitizer available here!"
- "Better safe than sorry!"
- "Use the alcohol!"
- "Clean hands, healthy body."
- "Clean hands protect your health."
- "Hand hygiene is a habit."
- "Prevent corona from spreading."
- "Infection control is in your hands."

- "Please decontaminate your hands."
- "Please clean your hands."

The next phrase was determined by the web server that kept track of the last three phrases used and circumvented them to avoid robots repeating the same message to a participant.

Additionally, a report page was available for the operator to analyze the experimental results for each day and each configuration, as well as to compare them, as exemplified by Figure 5.

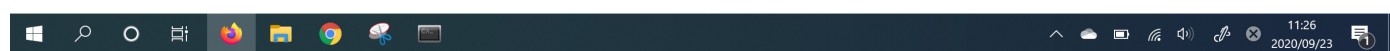

| Route (type 0) | Used Alcohol | Didn't use | Total |
|---|---|---|---|
| AB | 17 (10.97%) | 138 (89.03%) | 155 |
| AC | 1 (1.52%) | 65 (98.48%) | 66 |
| BA | 2 (2.04%) | 96 (97.96%) | 98 |
| CA | 4 (6.56%) | 57 (93.44%) | 61 |
| Going In | 18 (8.14%) | 203 (91.86%) | 221 |
| Going Out | 6 (3.77%) | 153 (96.23%) | 159 |
| Total | 24 (6.32%) | 356 (93.68%) | 380 |

**Figure 5.** Example of the user interface of the website to analyze data from each day.

*3.2. Methods*

Since the experiment involved humans, it received approval from the ethical committee of the Graduate School of Engineering Science, Osaka University.

3.2.1. Experiment Environment

In the corridor of the D entrance of the Engineering Science Building, two hand sanitizer dispensers, labeled 1 and 2, were placed over several days between 11:00 and 13:00, with a hand sanitizer dispenser at each end of the corridor. As shown in Figure 6, people going in (green, dashed) and out (magenta, dotted) of the building were considered part of the experiment. The route noted in blue was a free route for people who did not wish to participate and thus were not accounted for.

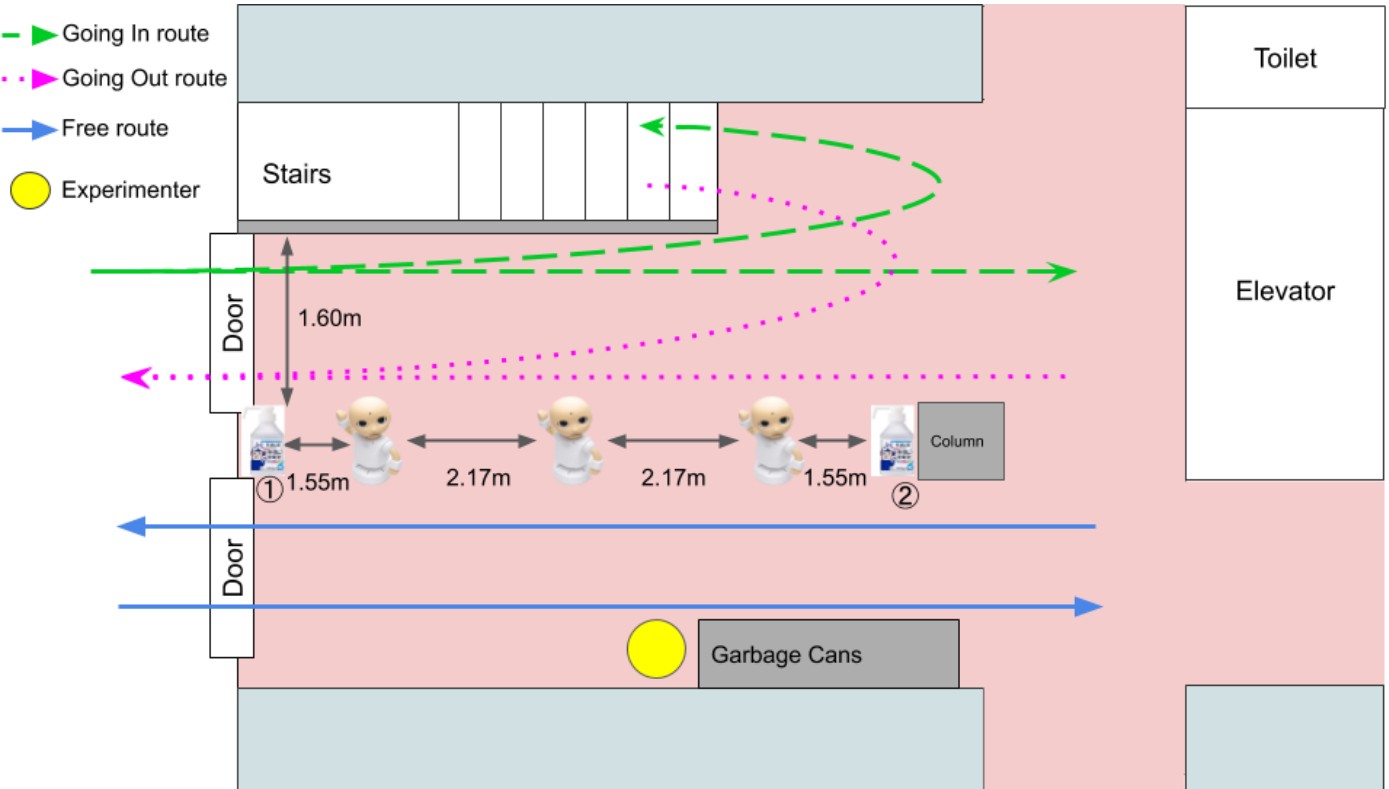

**Figure 6.** Map of the D entrance of the building of the Graduate School of Science Engineering in Osaka University with hand sanitizer dispensers and three robots in place. The distances noted are not to scale and were measured before the start of every experiment. The arrows represent the different routes a passerby might take during the experiment.

On alternating days, no robot, one robot, or three robots were placed between the hand sanitizer dispensers, promoting their use as a person passed by. As shown in the photos in Figure 7, the robots were positioned in such a way as to create a separation between people participating and not participating in the experiment. Furthermore, the robots had their backs toward the free route, which made them interact only with people who passed through the experiment route.

The robots were remotely controlled by the experimenter, who stood nearby the experiment area and supervised the whole experiment. To obtain more accurate data collection, video cameras were positioned under each robot and on top of the hand sanitizer to capture the feet and hands of passers-by walking through the experimental area and using the hand sanitizer. Furthermore, each day, the amount of alcohol used from each hand sanitizer was measured.

In the experiment, the operator appeared in the experimental space while any explanation about how the robot(s) was controlled was not explicitly given to the passers-by. Although it was possible that some or many of the passers-by noticed that the robot(s) was controlled by a human operator, it was not explicitly checked. Therefore, it is worth examining if or how much the reported effects depend on the passerby's cognition of the existence of the operator behind the robot(s).

It is worth noting that some participants might have noticed that a certain experiment was conducted in the corridor. In the current experiment, we could not check how many participants did so or examine how such an observation influenced the result. It is worth conducting further experiments with follow-up interviews to the participants to examine such a potential influence.

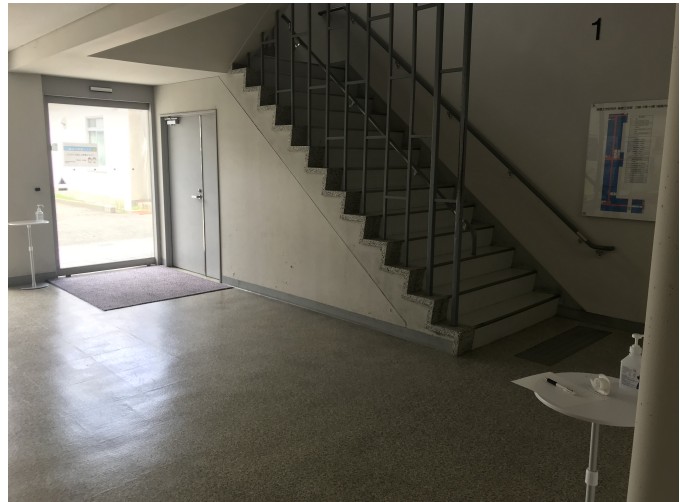

(**a**)

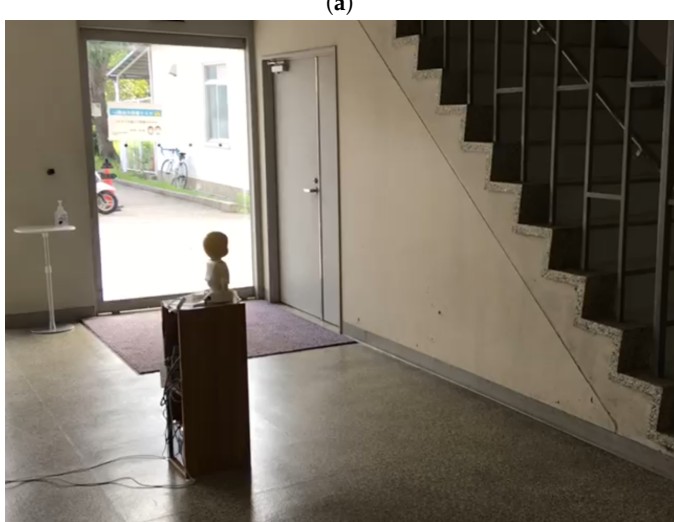

(**b**)

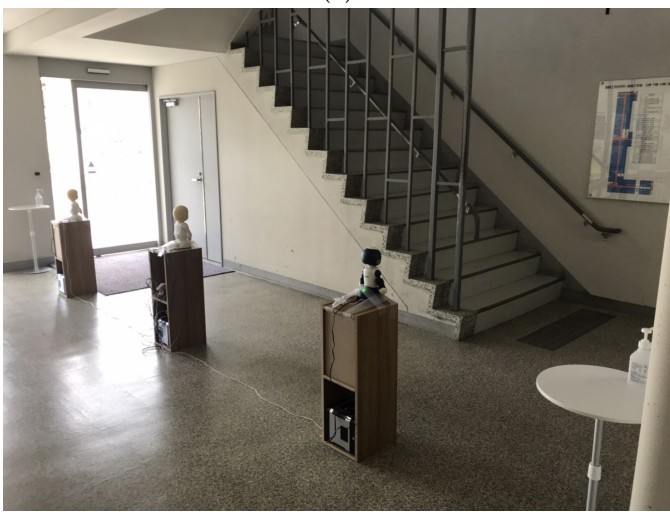

(**c**)

**Figure 7.** D entrance of the building of the Graduate School of Engineering Science in Osaka University during the experiment for the scenarios: (**a**) 0 robots; (**b**); 1 robot (**c**); 3 robots.

### 3.2.2. Subjects

This experiment was performed after some of the restrictions for COVID-19 were lifted from the university in July 2020. It is worth noting that during that period, undergraduate students were still advised not to go to campus, and in many locations hand sanitizer dispensers were already being placed to reduce the chance of contamination, such as at the entrances of buildings and laboratories. Additionally, the number of individuals infected in Japan gradually increased during the period. Table 1 includes the number of people infected each day, as reported by NHK (Japan Broadcasting Corporation), for future analysis, which may be a potential indicator for the people's anxiety for the spread of COVID-19 but were not directly analyzed in the current paper.

**Table 1.** Configuration of the experimental setup and the number of people found to be infected by COVID-19 on each day of the experiment.

| Day | Configuration | Number of People Found to be Infected in Japan |
|---|---|---|
| 21 July | 0 Robots | 632 |
| 22 July | 3 Robots | 795 |
| 27 July | 1 Robot | 597 |
| 28 July | 0 Robots | 981 |
| 29 July | 3 Robots | 1264 |
| 31 July | 1 Robot | 1579 |
| 3 August | 3 Robots | 959 |
| 5 August | 1 Robot | 1354 |
| 7 August | 0 Robots | 1605 |

After approval from the ethics council, an e-mail was sent to all building personnel to inform them that an experiment was being conducted at the D entrance of the building. Outside the building, as well as in the corridors inside the building that led to the D entrance, explanations about the experiment and a map indicating the experiment area and the free route were attached to the walls to let participants know and decide whether they would like to join the experiment.

Although there was no screening either before or after a person participated in the experiment, it was observed by the experimenter that the participants ranged from students to professors, school staff, and delivery personnel, both male and female.

### 3.2.3. Stimulus

As a participant entered the experiment area, a hand sanitizer dispenser was available at their side. As they walked toward the other side of the corridor, the robots (if any) said a phrase to promote the use of the hand sanitizer, as exemplified in Section 3.1.2. By the end of the corridor, the participant had another opportunity to use the hand sanitizer.

The manner in which the robots said their phrases depended on the experiment configuration. In the case of three robots, each robot said its phrase as the user passed by in front of the robot, for a total of three messages, while in the case of one robot, the single robot would say all three phrases in the same positions as in the three-robot configuration, as shown in Figure 8.

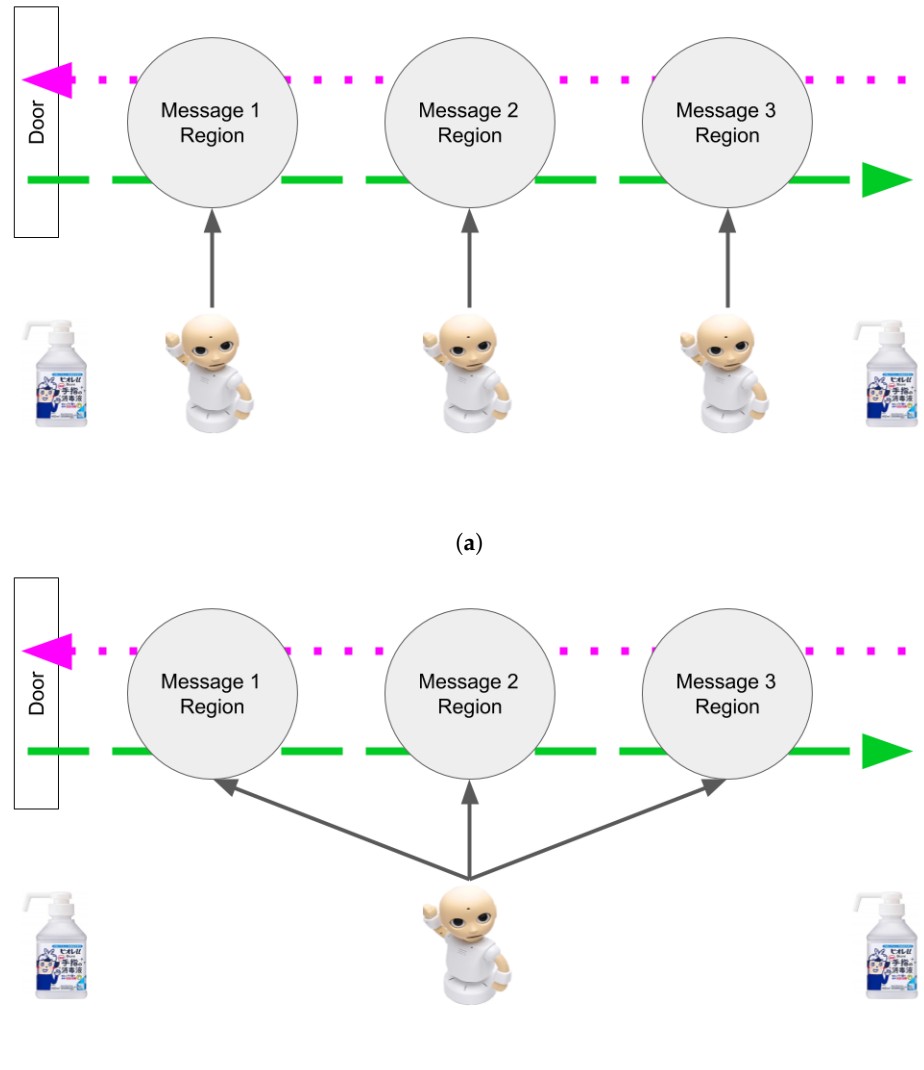

(**a**)

(**b**)

**Figure 8.** Diagram indicating the "Going In" (green, dashed) and "Going Out" routes (magenta, dotted), and the approximate region in which, after the participant entered, the operator activated the corresponding robot: (**a**) 3 robots (**b**); 1 robot . *Note:* Regardless of the configuration, the regions and timing of the message remained the same, with the only difference being that in the one-robot configuration, the solo robot was activated in all regions.

## 4. Results

During the nine days of the experiment, as indicated in Table 1, 2258 participants passed through the corridor and had the chance to use the hand sanitizer after passing by (or not) the robots. Figure 9 and Table 2 illustrate the proportion of participants that used the hand sanitizer in each scenario and the respective $\chi^2$ test *p*-values.

**Table 2.** Proportion of participants who used the hand sanitizer per configuration.

| Route | 0 Robots | 1 Robot | 3 Robots |
|---|---|---|---|
| Going In | 15% (74/509) | 36% (138/382) | 41% (10/264) |
| Going Out | 5% (20/402) | 20% (76/376) | 28% (91/325) |

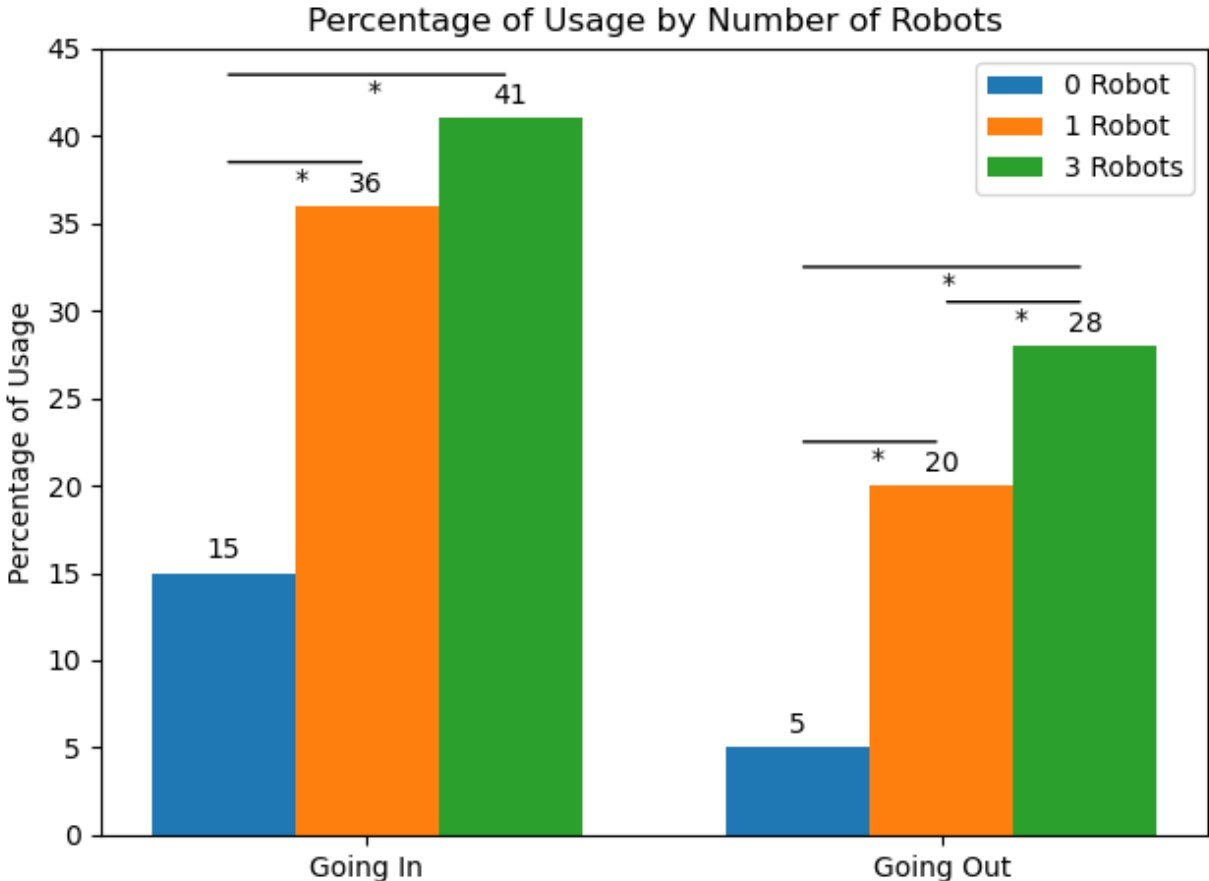

**Figure 9.** Proportion of participants who used hand sanitizers in the experiment grouped by route taken and configuration of the experiment. * $\chi^2$ test *p*-value < 0.05 after Holm correction

The $\chi^2$ test focusing on the data for the "Going In" route revealed a significant difference in the distribution ($\chi^2(2) = 81.73$, $p = 1.78 \times 10^{-18}$). We conducted a post hoc test on Z statistics with Holm correction to identify whether there were differences in the probability of using hand sanitizer among the various configurations. We found significant differences between zero robots and one robot ($Z = 7.489$, $p = 6.97 \times 10^{-14}$) and zero robots and three robots ($Z = 8.297$, $p \simeq 0$), while we did not find between and one robot and three robots ($Z = 1.327$, $p = 0.18442$).

The same analysis was conducted for the "Going Out" route. The $\chi^2(x)$ test revealed a significant difference in the distribution ($\chi^2(2) = 71.97$, $p = 2.35 \times 10^{-16}$). In the post hoc test with Holm correction, we found significant differences between the zero robot and one robot configurations ($Z = 6.457$, $p = 2.12 \times 10^{-10}$), as well as between zero robot and three robot configurations ($Z = 8.582$, $p = \simeq 0$). However, we did not find a significant difference between one robot and three robot configurations ($Z = 2.414$, $p = 0.01580$).

Between each day, the hand sanitizer bottles were weighed and the weights were recorded. The total usage of sanitizer in grams and the average per day per hand sanitizer dispenser are listed in Table 3, where "Bottle 1" was the one near the entrance and "Bottle 2" was the one near the corridor.

**Table 3.** Proportion of sanitizer usage (g) per configuration.

| Bottle | 0 Robot | | 1 Robot | | 3 Robot | |
|---|---|---|---|---|---|---|
| | Total | Average | Total | Average | Total | Average |
| 1 | 9 | 3 | 43 | 14.3 | 51 | 17 |
| 2 | 65 | 21.7 | 152 | 50.7 | 119 | 39.7 |

## 5. Discussion

From these results, it can be observed that there was a significant increase in usage from no robot to one robot, and even further to three robots, especially for people going out of the building. For participants going in, there was an observable increase; however, the number of participants was not high enough for statistical significance. It would be necessary to gather more subjects to produce a significant result. In addition, as reported in Table 1, the number of people found to be infected by COVID-19, which was a potentially confounding factor for the usage of hand sanitizer, varied among experimental days. Although there did not seem to be strong bias among conditions, it was difficult to quantify the confounding effect only with the current limited data. Therefore, in the future, it is worth running an experiment where such a confounding factor is controlled.

It is important to point out that people going in and going out had different tendencies as well, both with and without robots. One could argue that people would deem it more important to have their hands clean when going inside a closed environment, where they would be more likely to get an infection, while when they were going out, this fear would be reduced, and thus they would more easily neglect the hand sanitizer.

In the data on sanitizer usage, there is an observable increase in usage from no robot to many robots; however, it is worth noting that not only did each scenario have a different number of people passing by, but it was also observed that because the hand sanitizer was located in a public space, people who did not participate in the experiment also used it, and each person used a different amount to disinfect their hands, making it difficult to use these data for analysis.

According to [30,31], an argument presented by multiple sources is more persuasive than the same argument presented by a single source. While in this experiment it was possible to observe that people's behaviors changed more significantly with multiple robots than with a single robot, it is still not clear if they were truly persuaded to think that using the hand sanitizer was important as it was not possible to conduct any pre- or post-interviews with the subjects to confirm their bias, nor was there a method to keep track of their long-term behaviors.

Furthermore, because the experiment was performed in a public, open space, there was no control over which subjects participated in the experiment. A participant could pass by multiple times, or a certain profile of participants might have been more frequent than others. There was no feasible way to investigate this.

As noted in Section 2.1, this experiment only considered the situation in which the subject passively received the information from the robots. Although this is one factor that should be considered when designing a system, using this concept as an emergent theme, it is not yet possible to foresee all the different factors that influence the effectiveness of such a system.

## 6. Conclusions

In this study, a system was proposed that used not only one robot but, rather, multiple robots to spread a message to be delivered in sequence to a human, and it was theorized that this would increase the persuasiveness of the message.

Through a field experiment, it was possible to observe the increase in the promoted behavior by simply changing the number of robots used to deliver the messages. This experiment, however, had many limitations, and it is not possible to draw a deeper conclusion from only this study. Another study in a closed environment in which it would

be possible to evaluate the subject's acceptance of an argument in the long term, as well as control many of the variables that could not be controlled in this experiment, would be necessary to clarify such matters.

Overall, even though this concept is still not thoroughly tested, if designed carefully, it has the potential to significantly increase the effectiveness of an automated advertisement or public service announcement. It might seem like a small step toward making robots more acceptable and effective in society, but this small yet novel start could lead to more complex and novel systems in the future.

**Author Contributions:** conceptualization, M.I.T., K.O. and Y.Y.; methodology, M.I.T., K.O. and Y.Y; software, M.I.T.; validation, K.O. and Y.Y; formal analysis, M.I.T.; investigation, M.I.T.; resources, K.O., Y.Y. and H.I.; data curation, M.I.T.; writing—original draft preparation, M.I.T.; writing—review and editing, M.I.T.; visualization, M.I.T.; supervision, K.O., Y.Y. and H.I.; project administration, K.O., Y.Y. and H.I.; funding acquisition, Y.Y. and H.I. All authors have read and agreed to the published version of the manuscript.

**Funding:** This work was partially supported by JSPS KAKENHI Grant Number JP19H05691 and JST Moonshot R&D Grant Number JPMJPS2011.

**Institutional Review Board Statement:** The study was approved by the Ethics Committee of Graduate School of Engineering Science, Osaka University (R1-5-2, 10 July 2020).

**Informed Consent Statement:** Participant consent was waived due to decrease of data validity and quality. Instead, detailed explanation about the experiment including the passengers' rights and procedure to be excluded from the experiment was given both by broad posting and e-mail for all building personnel.

**Data Availability Statement:** The data presented in this study are available on request from the corresponding author.

**Conflicts of Interest:** The authors declare no conflict of interest. The funders had no role in the design of the study; in the collection, analyses, or interpretation of data; in the writing of the manuscript, or in the decision to publish the results.

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
