# Peer review of "Using Multiple Robots to Increase Suggestion Persuasiveness in Public Space"

_applsci, doi:10.3390/app11136080_

Round 1
Reviewer 1 Report
The authors propose a methodology of sequential persuasion, where multiple robots are used to deliver persuasive messages. The methodology was tested in a university campus, where it was observed the reaction of people to messages promoting the use of hand sanitizer, by considering three different conditions: no robot, one robot, and three robots. The robots are controlled by human operators.
The main goal of the paper is to explore whether such a sequential persuasion approach with n number of robots can more easily influence human behaviours. The work is written in an appropriate way, even if the introduction is too weak: the work is not well enough motivated and coherently with what is included in the introductory section. The originality of the work should be underlined by introducing in the paper a discussion of similar works in the literature because there are several works that propose the use of robots for persuasive purposes, but they are not examined in the text, nor is a comparison included. To mention only a few (although there are many other interesting ones in the literature) “Saunderson, Shane, and Goldie Nejat. "Investigating Strategies for Robot Persuasion in Social Human-Robot Interaction." IEEE Transactions on Cybernetics (2020).“ and “Ghazali, A. S., Ham, J., Barakova, E., & Markopoulos, P. (2018). The influence of social cues in persuasive social robots on psychological reactance and compliance. Computers in Human Behavior, 87, 58-65.”
The authors claim that the study focuses on improving the performance of small humanoid robots CommU in a public space. How the robot's performance has been improved? There is a comparison with a previous study?
For what concerns the results, the authors assert that they found significant differences between 1 Robot and 3 Robots for the "going in" scenario. Could you explain this statement with respect to the reported data?
The authors also discuss that it should be considered whether a human would perceive a system of robots as a group of multiple robots or as individuals. It this aspect has been evaluated?
Did people notice that the robots were remotely controlled?
Could you explain the header of table 1?
The authors should be more convincing about the originality and scientific robustness of the proposed approach for its publication in the journal.
Reviewer 2 Report
Please see the attached file

Reviewer 3 Report
This manuscript proposed a very interesting concept, namely ‘sequential persuasion’, and provided a rudimentary study on whether the number of robots would affect people’s tendency to use hand sanitizer. As the author honestly stated in their own word, “This experiment, however, had many limitations, and it is not possible to draw a deeper conclusion from only this study.” I have several questions regarding the overall research design and data analysis.
- The major concern is that this human subject-based observational research was performed during COVID, and the author have a table (Table 1) showing a surge in the number of COVID infections in Japan (the country where this research took place). I believe this would increase people’s tendency to use hand sanitizer. I believe this is a confounding factor that the authors need to discuss/address.
- Would the placement of video cameras affect human subjects’ behavior? If so, in which way?
- This work was built on the concept of ‘’sequential persuasion”, which is not new in psychology. I suggest the author to have a more expanded discussion reviewing relevant studies on the effectiveness of sequential persuasion.
- The data interpretation was not quite right. I would not call there is significant difference between 1 robot and 3 robots (p= 0.18442) on Page 8.
Round 2
Reviewer 1 Report
I would like to thank the authors for revising the paper according to the recommendations. Although I believe that the experimental study, as stated by the authors themselves, as well as the novelty of the proposed idea are not so strong to lead to relevant conclusions, it is appreciable that the study was conducted in such a difficult period during which it was quite impossible to carry out experiments in social robotics. I suggest in future work to further refine the methodology and to work more on the aspect of innovativeness and relevance to the scientific community.